# The Collective Influence of Intolerance of Uncertainty, Cognitive Test Anxiety, and Academic Self-Handicapping on Learner Outcomes: Evidence for a Process Model

**DOI:** 10.3390/bs14020096

**Published:** 2024-01-27

**Authors:** Jerrell C. Cassady, Addison Helsper, Quinton Quagliano

**Affiliations:** Department of Educational Psychology, Ball State University, 2000 W. University Ave., Muncie, IN 47303, USA; addison.helsper@bsu.edu (A.H.); quinton.quagliano@bsu.edu (Q.Q.)

**Keywords:** test anxiety, cognitive test anxiety, academic self-handicapping, intolerance of uncertainty, academic performance

## Abstract

Research on achievement emotions and behaviors has routinely demonstrated connections to student performance. This study approaches the work from a perspective of greater integration of multiple variables and examines potential nonlinear relationships among intolerance of uncertainty, cognitive test anxiety, academic self-handicapping, and student performance. Using structured equations modeling and generalized additive modeling, our results confirm better predictions of student performance when using both multivariate and curvilinear analyses. Optimal prediction of GPA was achieved when considering all three variables in conjunction through a serial mediation model. Specifically, the analyses demonstrated that intolerance of uncertainty tended to activate learners’ cognitive test anxiety, which increased the potential of engaging in academic self-handicapping, resulting in lower levels of GPA. The results are consistent with process models of test anxiety that identify the critical role of learners’ appraisals of evaluative stressors as well as the coping strategies employed when stressors are activated. Finally, curvilinear analyses confirmed that student GPA was related to academic self-handicapping and cognitive test anxiety in complex ways but largely demonstrated that as the levels of self-handicapping and/or cognitive test anxiety increased, performance declined. However, the degree of severity in reduced GPA was most severe for learners with elevations in both.

## 1. Introduction

The current investigation focuses on the power of intolerance of uncertainty (UNC) in predicting cognitive test anxiety (CTA) and, subsequently, the likelihood of engaging in academic self-handicapping behaviors and how those collections of tendencies impact student academic performance. While prior research often focuses on the direct influence of test anxiety on specific assessment events, this study focuses on learners’ grade point averages (GPAs) in order to examine the potential ongoing and cumulative negative influence that negative appraisals and coping strategies associated with test anxiety have on student performance in university settings. Based on the prior literature, we anticipate that high levels of intolerance of uncertainty will increase the level of cognitive test anxiety. Furthermore, we hypothesize that academic self-handicapping will interact with test anxiety to influence overall GPA such that learners with high levels of test anxiety will be more prone to academic self-handicapping, which will in turn lead to greater negative outcomes on GPA.

### 1.1. Cognitive Test Anxiety: Contributors and Consequences

Test anxiety has been a pervasive concern in educational settings for decades, and the evidence suggests that the levels of incidence are on the rise. Prevalence estimates have ranged from 15% to 70% of students over the years, with a trend toward higher rates in more recent studies [1]. There have also been increased levels of reported experience of anxiety as students progress through the educational cycle ranging from elementary school through university [2,3,4]. Most researchers identify two primary components of the broader notion of test anxiety, historically referred to as emotionality and worry [5]. “Emotionality” is more accurately described as the physiological component and is characterized by experiences such as shortness of breath, elevated heart rates, or nausea in response to evaluative contexts [6,7]. “Worry”, also referred to as cognitive test anxiety, captures the experiences students report involving cognitive processing inefficiencies, self-deprecating ruminations, cognitive distractions, and other impositions on working memory that deter optimal cognitive performance during test preparation or performance [1]. While individuals who report experiencing test anxiety generally experience both components, the data have routinely indicated that the cognitive test anxiety dimension is more directly linked to academic performance [8,9,10].

Conceptual models focused on how test anxiety arises and subsequently influences learners’ experiences, such as Pekrun’s Control–Value Theory and the S-REF model, converge in defining the key features to include (a) the learner’s appraisal of the event or task; (b) their estimation of the ability to assemble academic, emotional, and cognitive resources that will make the task manageable; and (c) the efficacy with which they effectively manage the demands of the task through effective self-regulatory and coping strategies [11,12,13]. When explaining the “cause” or “source” of elevated test anxiety, the learner’s appraisal of the event is a critical feature. Therefore, when alerted to an impending evaluation, the learner forms an appraisal of the potential that they can effectively manage and succeed on the task based on the interplay between (a) perceived task requirements, (b) available supports and resources (including time), and (c) confidence in their abilities [11,12]. The learner uses this appraisal to determine the degree of perceived likelihood for success or threat. Historically, test anxiety research and theory have demonstrated that as their appraisal for succeeding in the evaluation increases, the level of test anxiety decreases [14]. While this linear trend has been presented repeatedly, recent research has proposed that a more nuanced curvilinear relationship enables a more accurate fit to the data. Specifically, students who were more uncertain about the outcome of a forthcoming assessment reported the highest levels of cognitive test anxiety; learners who were confident they would perform either poorly or well reported significantly lower levels of test anxiety [15].

In a related line of research, when test anxiety is elevated due to learners’ appraisal of threats imposed in academic settings, a commensurate decrease in performance has been routinely observed. However, researchers also demonstrated that levels of test anxiety and performance may also be more accurately captured with a curvilinear relationship. Specifically, research has suggested that a “little” test anxiety may provide a facilitative influence on performance by spurring an activating motivation to prepare for upcoming tests, but when the level of test anxiety reaches a critical point, the deleterious effects follow rapidly [16]. The negative outcomes for learners with heightened test anxiety on performance measures have been demonstrated across all forms of assessment, ranging from non-evaluative practice exams to high-stakes assessments and cumulative GPA [14,17,18]. The traditional representation of how test anxiety influenced these negative outcomes in performance was to focus on cognitive distractions or interference during testing events [19,20]. However, contemporary research on test anxiety has expanded to track the influence of test anxiety on beliefs and behaviors during three phases in the learning–testing cycle that interfere with optimal performance. Research on these phases has demonstrated that learners with test anxiety report (a) higher rates of a perceived threat of forthcoming assessments, less effective study routines and habits, and higher rates of maladaptive coping strategies during the test preparation phase; (b) suboptimal test-taking strategies during the test performance phase; and (c) higher prevalence of hopelessness orientations following exams (which predispose the learner to heightened perceived threat for the next testing event in a vicious cycle [9,21]).

### 1.2. Intolerance of Uncertainty

As proposed by appraisal-focused models of test anxiety [11,12,13], the learners’ degree of uncertainty about the outcome of the evaluative situation is a key contributor to the manifestation of anxiety. Research in anxiety disorders over the past two decades has demonstrated that uncertainty itself may not be the main contributor to anxiousness but rather the individual’s “intolerance” of that feeling of uncertainty. Intolerance of uncertainty (UNC) has been defined in various ways, but Carleton’s [22] representation captures the key elements as a dispositional construct that focuses on the inability to accept the absence of clear or sufficient information in situational contexts. Extensions of this definition often focus on the suggestion that this discomfort over unpredictability points to individuals seeing uncertainty as either threatening or unbearable, even when negative outcomes are unlikely [23], and those with high levels of UNC are prone to inherently judge ambiguous situations as threatening [24]. UNC has been identified as a critical feature in models of Generalized Anxiety Disorder (GAD) [25], wherein UNC elevates the psychological threat posed by “what if” questions posed in situational contexts. As answers to such queries become less clear, pervasive worry ensues, which increases the potential for an anxious response. In that context, the anxious individual is likely to engage in cognitive avoidance strategies and develop less accurate and manageable representations of the situation leading to the worrying event [22,25,26].

While much of the work on UNC has centered on clinical anxiety disorders such as GAD [27,28], obsessive–compulsive disorder [29], and social anxiety disorder [30], Carleton et al. [31] argued that UNC appears to be a central feature in situation-specific anxiety responses. Exploring the relationships between UNC and test anxiety has confirmed the conjecture. For instance, Putwain and Pescod [32] demonstrated that a key feature in test anxiety reduction training was that uncertainty control mediated the cognitive dimensions of test anxiety over time. In their intervention, students learned strategies to control negative thoughts, prepare for exams, and decrease ruminating over failure. The program was demonstrated to be effective in reducing test anxiety, but that reduction in test anxiety was mediated by changes in students’ control over uncertainty. Putwain and Pescod proposed that test anxiety declined because learners perceived they had greater control over positive outcomes in a forthcoming evaluation (thereby reducing uncertainty). Similarly, in a predictive modeling study, Tang [33] confirmed that UNC and CTA are highly correlated and subsequently linked to academic performance. Tang also demonstrated that resilience was inversely related to CTA and UNC, but the analyses did not directly assess whether resilience mitigated the negative academic outcomes predicted by CTA and UNC.

Pekrun’s Control–Value Theory of Achievement Emotions [12] effectively explains the collective work on intolerance of uncertainty as viewed from both clinical and pre-clinical lenses. In his model, the critical situation arises when a learner has a situational outcome expectancy that anticipates failure is a possibility and that such a failure poses a threat to the learner. Once this representation has been established, the learner makes a predictive determination of whether the resources at their disposal (e.g., study skills, cognitive abilities, time, emotion regulation skills, supportive structures) are sufficient to ensure success in the current context [12]. When the prediction of success by the learner is “uncertain” (or beyond the individual’s control), anxiety is most likely to arise, and the distress is predicted to be higher than if the learner is confident that the outcome will be unsuccessful [15,34]. While research in clinical (e.g., GAD) and pre-clinical anxieties (e.g., CTA) both identify anxiety as the affective response when the outcome is uncertain, the terms used in these two domains of anxiety research tend to differ when describing the emotional responses arising from certain failure and success. Within clinical anxiety discussions, predicted failure is referred to as fear, while success engenders feelings of calm [22]. By contrast, Pekrun classifies certain failures as sparking feelings of hopelessness, while predicted success for upcoming tests sparks feelings of relief [12].

Following test anxiety models, an anxious response often activates a coping response that is intended to mitigate the negative affective situation [11,12,13]. The research on UNC and CTA have both demonstrated that avoidance strategies are common coping responses when the anxiety response is strong or accompanied by uncertainty [35]. Viewed from an emotion regulation perspective, learners experiencing high levels of test anxiety who cope through withdrawal or avoidance are effectively achieving the goal of reducing anxiety (at least in the moment). However, the temporary release from negative affective experience ultimately exacerbates the likelihood of subsequent test failure, which is why this approach to coping is often identified as engaging in academic self-handicapping.

### 1.3. Academic Self-Handicapping

Academic self-handicapping is defined as the tendency to impose or identify impediments or barriers that restrict optimal performance on academic tasks [36]. The critical difference between academic self-handicapping (ASH) and other general attributional claims regarding performance failures is that ASH requires a priori identification of the factor that will lead to reduced potential for success. The standard explanation for why learners engage in self-handicapping behaviors is through a form of ego defense, providing a plausible explanation for suboptimal performance that is externalized and helps the learner maintain their academic identity [37,38]. This ego defense mechanistic strategy is the central assertion in Schwinger et al.’s proposed model for antecedent contributors to ASH [39]. Their model identifies personality and situational factors that have been demonstrated in the literature to be associated with ASH, with all the proposed antecedents (e.g., neuroticism, test anxiety, fear of failure) flowing through a threat to self-esteem. However, as test anxiety models focused on appraisal and coping would propose, ASH may also arise due to the learner’s attempt to escape negative affective stimuli sparked by anxiousness over forthcoming evaluation [11,12,13].

In a meta-analysis examining relationships between ASH and academic performance, a moderate and durable effect was reported for cumulative measures, such as GPA and classroom test performances [40]. Evidence focused on the progressive influence of ASH on tests over the course of a semester [38], as well as cumulative GPA [41], indicate that while engaging in academic self-handicapping behaviors may provide an initial benefit to ego defense, a progressive, cumulative negative effect is observed. The durability of the link between ASH and academic performance, in conjunction with mixed effects with contributing factors seen to be related to ASH (e.g., personality factors, goal strategies, motivational perspectives), has led some researchers to suggest that ASH is a key mechanism through which performance declines are instantiated [39,41]. While the general construct of ASH is the usual focus of theoretical discussions, there is a clear delineation in the literature between two primary forms of self-handicapping. Behavioral self-handicapping comprises self-handicapping activities that are knowingly incompatible with successful performance on a forthcoming task, resulting in a form of self-sabotage [42]. Such strategies may involve choosing to go to a party rather than study the night before a test or openly defying course expectations to submit work or attend class. Conversely, “claimed” self-handicapping is an explanation or justification for lower performance that is known and identified prior to the academic activity but is not a voluntary behavior. One of the more common examples of claimed academic self-handicapping is test anxiety, where the learner predicts that their performance will be hampered by their test anxiety [36]. When used as an academic self-handicap, test anxiety serves the role of preserving the student’s academic identity by identifying an a priori rationale for unfavorable performance that is not directly related to their ability. Other forms of claimed self-handicapping may involve identifying an illness ahead of a due date for an academic task or perhaps a stereotype threat [39,42].

Most researchers maintain that claimed self-handicapping conditions are relevant, but many concede that some instances of claimed self-handicapping may not be truthful [36]. Our attention to self-handicapping is focused entirely on the behavioral component because it fits in the overall process models of test anxiety more directly as a clear coping response that may interact with test anxiety (which we assert is a distinct construct). Evidence for the distinction between test anxiety and self-handicapping was provided by Putwain [35] in a study exploring self-handicapping and the cognitive component of test anxiety (worry), illustrating that ASH was only reasonably related to test anxiety and performance in a path model where the influence was mediated fully through the intervening variable “beliefs about control.” Putwain’s studies [32,35] demonstrated the importance of multiple variables simultaneously influencing the understanding of the sources and impacts of test anxiety and provided a key inspiration to this study design. Consistent with those studies, our work is interested in exploring antecedent variables likely to instantiate test anxiety in learners through their appraisal of the degree of threat in impending evaluative situations, as well as examining the collective influence of these variables on academic performance. Where our study differs most directly is in the hypothesized relationships among ASH, CTA, and academic performance.

### 1.4. Current Study

The purpose of this study was to examine the relationships among learners’ cognitive test anxiety, intolerance of uncertainty, academic self-handicapping, and student grades with a specific focus on (a) process models for achievement emotions, (b) multivariate solutions that examined combined and individual influences of the variables, and (c) reviewed the relationships for both linear and curvilinear influences. Our primary expectations were that intolerance of uncertainty would have a strong influence on the level of test anxiety, test anxiety would subsequently be associated with student performance as well as academic self-handicapping, and that self-handicapping would moderate or mediate the relationship between test anxiety and academic performance. Finally, consistent with recent evidence on relationships between test anxiety and performance, we anticipated that a curvilinear relationship between test anxiety and performance was likely.

## 2. Materials and Methods

### 2.1. Participants

The individuals were undergraduate students recruited from a midsized public university located in the Midwestern United States. Voluntary participation in this study was one of several options for course credit in multiple courses delivered in a school of Education across 6 academic semesters, consistent with the protocol approved by the university institutional review board. After providing consent to participate, participants completed an online survey through the Qualtrics platform.

Our final sample consisted of 853 students who completed the survey, which was available from 2021 to 2023. An initial data set of 943 participants was reduced to this sample size after (a) 44 participants were removed as multivariate outliers from a Mahalanobis distance test (df = 42, x^2^ = 76.3, *p* = 0.001) and (b) an additional 46 students were removed via listwise deletion due to no reported cumulative GPA. Listwise deletion was selected as the appropriate method for removing participants lacking the single-item outcome variable in our SEM tests, as complete data on this crucial variable are essential for accurate model estimation and the validity of our results. All other missing values of the final sample were subject to imputation. Table 1 reflects the demographic breakdown of our final sample.

### 2.2. Measures

Cognitive Test Anxiety. Participants completed the cognitive test anxiety scale—2nd edition (CTAS-2), which is the latest revised form of assessment based on the original cognitive test anxiety scale [1,8]. The measure includes 24 items assessing learners’ reactions to impending evaluative situations using a 4-point response scale (1 = Not at all like me; 4 = Very much like me; full list of items available at Cassady, 1). Multiple validation studies have demonstrated that the CTAS-2 shares concurrent validity with other measures of academic anxiety and is represented as a form of pre-clinical anxiety that is predictive of the respectively broader constructs of academic anxiety and generalized neuroticism. The scale has been translated for use in over 20 countries and is repeatedly found to have high internal consistency (Cronbach’s α = 0.96, McDonald’s ω = 0.96) [1].

Intolerance of Uncertainty. The shortened version of the intolerance of uncertainty (12-item version; IUS-12) [22,23] was used to estimate the overall degree of intolerance of uncertainty reported by the participants. Responses to the 12 items are provided on a 5-point Likert-type scale (1—*Not at all characteristic of me*; 5—*Entirely characteristic of me*). Evaluation of the factor structure of the 12-item version verified an underlying two-factor structure, with 7 items measuring Prospective Anxiety and 5 items identifying inhibitory anxiety. The prospective factor centers on items that address the participants’ determination that uncertainty is unacceptable and should be avoided. The inhibitory subscale highlights participants’ inability to act when faced with uncertainty [22]. For all analyses, we focused on the total combined scale score, which was demonstrated to have high concurrent validity with measures of anxiety, the original 27-item version of the IUS, and had a high internal consistency (alpha = 0.91) [22].

Academic Self-Handicapping. A 6-item version of the academic self-handicapping scale (ASHS) [36,43] was used to assess the students’ self-described use of self-handicapping strategies in response to academic stressors. The selection of this approach to measuring academic self-handicapping was supported by Schwinger et al.’s meta-analytic findings that concluded the ASHS provided more utility value in studies focused on academic performance outcomes [40]. Proposed advantages for assessing self-handicapping provided by the ASHS include clear statements of the self-handicapping behavior, rationales for engaging in the behavior, and timing of the behavior prior to the academic failure [36,40]. As such, the ASHS measures “behavioral” self-handicapping rather than “claimed” self-handicapping.

### 2.3. Data Analysis

#### 2.3.1. Structural Equation Modeling

To address the research questions of interest, four separate structural equation models (SEMs) were fit to the data. These models were based on the theoretical framework that intolerance of uncertainty (UNC), cognitive test anxiety (CTA), and academic self-handicapping (ASH) influence long-term academic performance, measured by grade point average (GPA). In the development of our methodological approach, particularly our step-by-step structural equation modeling (SEM) process, we draw upon established practices and advancements in the field, such as the suggested approach provided by Suhr [44]. In particular, our iterative process detailed in this section highlights the final four suggested steps, including parameter estimation (CFA), assessing model fit (Model 1), model modification (Models 2–4), and finally, the interpretation and presentation of results. When looking at multiple models of an SEM, these model modifications can be interpreted much in the same way as post hoc comparisons act for an ANOVA. 

The models explored included a mediation model with CTA as the mediator between UNC and GPA (Model 1), an extended moderation (Model 2), a subsequent mediation model (Model 3) incorporating ASH as an additional moderator/mediator for CTA, and a serial mediation model examining the combined mediating effect of CTA and ASH (Model 4). The optimal model was determined based on fit indices and the subsequent cut-off values recommended by Kline [45], such as the comparative fit index (CFI) of >0.90, Tucker–Lewis Index (TLI) of >0.90, root mean square error of approximation (RMSEA) of 0.05, and standardized root mean square residual (SRMR) < 0.1. Missing data were determined to be missing completely at random. Prior to model testing, data showed a strong skew in multivariate normality. Robust fit measures and extraction methods were utilized to account for the abnormal distribution of observed variables.

Once an optimal SEM model was identified through confirmatory factor analysis (CFA), the focus shifted to examining the direct and indirect relationships between the variables. All model parameter estimation was carried out using the lavaan package in R, with a maximum likelihood estimation method (MLM).

#### 2.3.2. Nonlinear Models

To explore potential nonlinear relationships among cognitive test anxiety (CTA), academic self-handicapping (ASH), and student GPA, we employed generalized additive models (GAMs) [46]. GAMs are particularly suited for identifying and illustrating nonlinear relationships in data, allowing for more nuanced interpretations than linear models. Nonlinear models were fit using the gam library in the R software environment, version 4.1.3 [47]. The fitting process involved iteratively weighted least squares with a thin plate smoothing spline, allowing for both smoothed (nonlinear) and parametric (linear) terms for each independent variable. An F-test was employed for each term to determine statistical significance, with an alpha level set at 0.05.

## 3. Results

### 3.1. Latent Variable Confirmatory Factor Analysis

Confirmatory factor analysis (CFA) was conducted to evaluate the model fit for the three primary latent variables. The CFA model included three latent variables: intolerance of uncertainty (UNC), cognitive test anxiety (CTA), and academic self-handicapping (ASH), along with the observed outcome variable of cumulative grade point average (GPA). To enhance the model fit and account for previously established correlations, modification indices were carefully examined. Indices suggesting values above 15 and subsequently above 10 were considered for model adjustments. These modifications primarily involved adding covariances among the latent variables while ensuring that direct relationships and covariances with the outcome variable were excluded. This exclusion was critical to maintaining the integrity of the direct and indirect paths to be analyzed in the subsequent SEM models. The CFA was performed using the lavaan package in R, with a maximum likelihood estimation method (MLM). The analysis included a total of 853 observations, with the model comprising 215 parameters. The CFA and subsequent SEM models demonstrated significant deviations from a perfect fit (*p* < 0.001). However, it is worth noting that the chi-square test, traditionally used as a hypothesis test for SEM models, may lose power when dealing with dichotomous or non-normally distributed indicators [48]. Additionally, it can be overly sensitive to invariance issues between observed and predicted models in larger sample sizes [49]. Therefore, to assess model fit, we relied on fit statistics rather than the chi-square hypothesis test. For example, the robust comparative fit index (CFI) and robust Tucker–Lewis index (TLI) were 0.960 and 0.950, respectively, suggesting an acceptable fit to the data. The root mean square error of approximation (RMSEA) was 0.039, and RMSEA values were well below the conventional threshold of 0.05, indicating a good fit. The standardized root mean square residual (SRMR) was 0.047, further confirming the adequacy of the model fit. All suggest an adequate fit to the data prior to the addition of relevant direct and indirect relationships to be tested by the SEM [45].

### 3.2. Structural Equation Modeling: Mediation and Serial Mediation Models

#### 3.2.1. Model 1: CTA Mediating UNC and GPA

The first SEM was conducted to examine the mediating role of cognitive test anxiety (CTA) in the relationship between intolerance of uncertainty (UNC) and grade point average (GPA). The (CFI) and (TLI) were 0.959 and 0.950, respectively, suggesting an acceptable fit to the data. Additionally, the RMSEA was 0.040, and the SRMR was 0.048, both of which indicate a satisfactory model fit. Parameter estimates revealed significant paths: UNC positively predicted CTA (β = 0.627, *p* < 0.001), and CTA negatively predicted GPA (β = −0.194, *p* < 0.001). The indirect (mediation) effect of UNC on GPA through CTA was significant (β = −0.122, *p* < 0.001). Factor loadings for the 24 individual items of CTA ranged from 0.508 to 0.921, the 12 loadings for UNC ranged from 0.431 to 1.038, and the 6-item factor loadings for ASH ranged from 0.691 to 1.024.

#### 3.2.2. Model 2: ASH Moderating CTA and GPA

Model 2 sought to extend the SEM framework by adding academic self-handicapping (ASH) as a subsequent moderator in the relationship between CTA and GPA. The CFI and TLI values were 0.276 and 0.259, respectively, suggesting a poor fit to the data. The RMSEA was 0.164, and the SRMR was 0.229, both outside the acceptable limits for a good model fit. Given the inability to fit a moderation effect model, we turned our attention to exploring the possibility that ASH was a mediating variable in the model.

#### 3.2.3. Model 3: ASH Mediating CTA and GPA

Model 3 extended the SEM framework by adding academic self-handicapping (ASH) as a subsequent mediator in the relationship between CTA and GPA. The CFI and TLI values were 0.959 and 0.950, respectively, suggesting an acceptable fit. The RMSEA was 0.040, and the SRMR was 0.048, both within acceptable limits for a good model fit. Significant paths in the model included UNC predicting CTA (β = 0.627, *p* < 0.001), CTA predicting ASH (β = 0.227, *p* < 0.001), and CTA negatively predicting GPA (β = −0.194, *p* < 0.001). Additionally, ASH negatively predicted GPA (β = −0.353, *p* < 0.001). The model identified two significant indirect (mediation) effects, UNC on GPA through CTA (β = −0.122, *p* < 0.001), and a second indirect effect, CTA on GPA through ASH (β = 0.068, *p* < 0.001). Model 3’s SEM analysis confirmed the hypothesized mediating roles of CTA and ASH in the relationship between UNC and GPA, with both indirect paths showing significant effects.

#### 3.2.4. Model 4: Serial Mediation Model with ASH and CTA

The final model in our analysis tested the serial mediation effect of CTA and ASH on the relationship between UNC and GPA. See Figure 1 for the visualization of the final model and relevant coefficients. The model demonstrated an acceptable fit with a robust CFI of 0.957 and a robust TLI of 0.947. The RMSEA was 0.040, and the SRMR was 0.047, both indicating a good model fit.

All direct and indirect relationships in the model were statistically significant. The indirect effect of UNC on GPA through CTA (ind1) was −0.117 (*p* < 0.001), indicating a significant mediation effect. The indirect effect of ASH on the CTA-GPA relationship (ind2) was 0.082 (*p* < 0.001), and the serial mediation effect of CTA and ASH on the UNC-GPA relationship (ind3) was −0.089 (*p* < 0.001). The R-squared for GPA in this model was considerably stronger than in Models 1 and 3, suggesting the serial mediation solution was optimal (see Table 2).

### 3.3. Nonlinear Modeling

While the exploration of the full model was the area of primary interest to see the collective and interactive effects of all three constructs on GPA, our continued preference for expanding attention in the field to curvilinear effects between variables that have been traditionally studied only through linear effects prompted a review of the relationships among the primary variables. In particular, we tested for nonlinear patterns for (a) cognitive test anxiety and GPA; (b) academic self-handicapping and GPA; and (c) a three-way relationship between test anxiety, academic self-handicapping, and GPA. To test for potential curvilinear relationships, a generalized additive model (GAM) was used. A GAM is a statistical technique that, through the introduction of smoothing functions, enables seeing more complex relationships past traditional linear models through nonlinear relationships between the independent and dependent variables [50].

#### 3.3.1. Cognitive Test Anxiety and GPA

The results revealed that the nonlinear relationship between test anxiety and GPA was statistically significant (F(3.678, 853) = 13.01, *p* < 0.0000; Figure 2). The *y*-axis reflects smoothed GPA, and the *x*-axis displays CTAS2. In addition to the relationship between the two variables (represented by the line), the sample size at each CTAS2 value is displayed in the rug plot on the *x*-axis. This graph shows that as the level of cognitive test anxiety increases, student GPA decreases with a notable curvilinear trend, with the steepest decline in GPA seen in scores ranging between 1.5 and 3.25. At those “tails” of very low and very high test anxiety, the GPA values plateau (and the sample sizes also drop considerably, accounting for the wide confidence intervals).

Of particular interest is the relationship between the mean-centered GPA and value on the CTAS-2. The values expressed on the *x*-axis are representations of individual item averages. Prior work identifying “levels” of cognitive test anxiety with the CTAS-2 has produced cut scores for individuals with low, moderate, and high levels of cognitive test anxiety [1]. To identify the relationship between those pre-determined levels and GPA, vertical lines associated with the cut score values are superimposed on the GAM chart. In essence, the chart demonstrates that the traditional and simple “linear” result for test anxiety and GPA is clearly replicated for the “moderate cognitive test anxiety” group such that as the level of CTA for students in the moderate range increases, a stead and commensurate drop in GPA is observed. The significantly less dramatic slopes observed for the “high” and “low” groups suggest that when in the extreme ends of test anxiety grouping a ceiling effect, there is more homogeneity due to essentially floor and ceiling effects due to the restricted range of possible GPAs. However, it is also clear that the students in the high and low groups of CTA maintain significantly different GPA profiles.

#### 3.3.2. Academic Self-Handicapping and GPA

The results revealed that the nonlinear relationship between academic self-handicapping (ASH) and GPA was statistically significant (F(8.955, 853) = 6.434, *p* < 0.00001; Figure 3). The *y*-axis reflects smoothed GPA, and the *x*-axis displays ASH. In addition to the relationship between the two variables (represented by the line), the sample size at each ASH value is displayed in the rug plot on the *x*-axis. This visualization aids in understanding the complex, nonlinear influence of academic self-handicapping on GPA and illustrates the dramatically low level of endorsed behaviors.

As indicated in the figure, the vast majority of respondents reported an average value of less than 2.0 on the items (response range 1–5). The limited number of respondents to the right half of the graph (i.e., ASH > 3) provided lower degrees of certainty, limiting interpretability. However, the trend seen for students in the left half of the graph illustrates that students who infrequently use ASH perform optimally in GPA. The bell-shaped relationship shown for values ranging from an average of 1 to 2 on ASH items suggests that students who report using one or perhaps two academic self-handicapping behaviors on an infrequent basis show a slight gain in GPA over those who claim to never engage in any academic self-handicapping activities. However, once learners report using more than two academic self-handicapping behaviors (infrequently) or one behavior regularly (resulting in an average value of 1.5 on this chart), their GPA starts to decline rapidly.

#### 3.3.3. Cognitive Test Anxiety × Academic Self-Handicapping on GPA

Our final curvilinear exploration was focused on the interactive effects of CTA and ASH on GPA. The results demonstrated a significant interaction between academic self-handicapping (ASH) and test anxiety (CTAS2) in predicting student GPA (F(6.566, 853) = 11.35, *p* < 0.00001; Figure 4). The model’s adjusted R-squared value of 0.077 and the deviance explained at 8.31% indicate that the combined influence of ASH and CTAS2 on GPA is both significant and complex.

The dots displayed in Figure 4 identify the actual data points for students’ responses to the CTA and ASH surveys. Areas with a high concentration of black dots (or darker dots) indicate a high density of responses. The contour lines represent GPA values, with the curvilinear path identifying the interactive levels of CTA and ASH. The gradient of the lines from red to blue represents a spectrum of GPA levels, with one end indicating lower GPAs and the other end indicating higher GPAs.

Examining the GPA bands with respect to the collective influence of CTA and ASH identifies several interesting trends. First, the highest GPA band is tightly clustered in the lower left corner of the figure, indicating that these students report low levels of both CTA and ASH. The next band of GPA shows a roughly linear trend, indicating that relatively successful students are likely to report only a low to moderate level of test anxiety or a small degree of academic self-handicapping. However, this GPA group seldom reports having elevated levels of both variables. As the GPA lines become more red (lower GPA), a consistent trend is observed. As the GPA values get lower (illustrated both by color and also by the lines progressively further right on the figure), the level of CTA becomes considerably higher, and the level of ASH also increases. The two GPA band lines with a similar bend indicate that it is possible for students to fall into this lower range of GPA with low levels of ASH, but only if their reported level of CTA is in the “high” group (as indicated by scores above 2.8 on this figure). As the level of ASH increases for these two groups, the degree of CTA drops precipitously. For those students with both high CTA and high ASH (the reddest and right-side lines), their GPA drops significantly. As illustrated by the density distributions, these groups also represent a considerably small portion of the sample.

## 4. Discussion

The results of this study exploring relationships among intolerance of uncertainty (UNC), cognitive test anxiety (CTA), academic self-handicapping (ASH), and cumulative GPA provided support for process models identifying the importance of observing learners’ beliefs and behaviors related to academic evaluation situations to more completely recognize the interrelations among these variables. Consistent with existing models, the results from our study confirmed that the ability to predict students’ cumulative GPAs became considerably more accurate as the relationship among the primary variables in question was ordered in line with models that identify the importance of appraisals and coping responses during academic tasks that may pose stress via perceived threat of failure [9,11,12,13]. While several prior studies have documented the relationships among specific elements under investigation in this study [32,33,40], this study builds upon Putwain’s breakthrough study, exploring multiple influences [35] by adjusting the location of the constructs as well as examining a different population. We see three primary contributions to the literature that warrant further discussion. First, the optimal predictive model for GPA operated such that individuals with high levels of UNC were more likely to have heightened CTA, which in turn increased the probability of self-reported ASH and subsequent declines in GPA. Second, the direct relationships between CTA and GPA, as well as ASH and GPA, demonstrated distinctly curvilinear trends that highlight the importance of examining significant results from linear analyses more completely to identify underlying nuances. Third, the ability to examine the curvilinear relationship between the interaction of cognitive test anxiety and academic self-handicapping as operating on student GPA is a considerable breakthrough afforded by new analytic and data visualization techniques. Moreover, the results identified in that analysis help explain the pattern of results identified in the separate SEM models.

### 4.1. Process Model for Achievement Emotions and Performance

Consistent with prior models that examine achievement emotions such as test anxiety and self-handicapping, our results demonstrated a clear and significant relationship among the four primary variables under investigation. The data demonstrated that simple models (e.g., Model 1) focused on individual mediation effects provided a reasonable (but small) effect and could be used to explain some of the variance in GPA. However, as the model became more complex and complete, the solution accounted for notably higher levels of variance in GPA. We interpret our final solution as supporting process models for emotion and performance [11,12,13]. Specifically, our results suggest that the process follows a progressive emotion regulation and self-regulated learning approach that highlights the importance of learners’ appraisals of the potential threat imposed by an impending evaluation, the instigation of test anxiety as a reaction when they determine their resources or skills are likely insufficient to lead to success, and the considerable negative consequences that result when they implement coping strategies that are contrary to effective test preparation strategies. The improved value of the serial mediation model (Model 4) over the prior solutions demonstrated that while each component of the model (UNC and CTA or CTA and ASH) has value, it was the collective set of beliefs and behaviors that was most successful in identifying student GPA.

A review of the entire model (Figure 1) illustrates that there are nuances among the variables, which are largely a consequence of students’ differential experiences with the three predictor variables. For instance, while UNC is most directly impactful on GPA in a negative relationship, as mediated through CTA and CTA followed by ASH (2 complementary mediated pathways), there is also a small but significant positive direct effect on GPA. As such, there is some evidence in this model that UNC for students who do not have heightened levels of test anxiety or self-handicapping behaviors may have a slight facilitative effect. This could be articulated as learners who are uneasy with limited predictive accuracy in knowing the outcome of tests (perhaps because they are in difficult courses) but do not suffer from test anxiety may be sparked to engage in more preparation activities to alleviate that level of uncertainty. This positive coping response as a reaction to a primary appraisal of the “threat” imposed by the test is more likely from an individual whose secondary appraisal (where their ability to meet the challenge imposed by that threat is determined) concludes they have access to the necessary time and resources to be successful [11].

For those learners who have less confidence in their ability to succeed in the task at hand, cognitive test anxiety may be activated. Models 1, 3, and 4 all confirm that CTA mediates the relationship between UNC and GPA. In essence, these results indicate that the majority of the influence of UNC operates through CTA (and likely other related academic anxieties), supporting prior work suggesting UNC is a central component of situation-specific anxieties [31,32]. This study provided new information showing that once UNC has activated CTA for learners with appraisals that maintain a risk of failure, the coping strategies that are employed become a critical feature in determining eventual GPA. For those learners with high UNC and CTA who enact self-sabotaging or self-handicapping behaviors (procrastination, withdrawal, avoidance), the impact on GPA is notable. Reviewing this from the perspective of a process model for achievement emotions, the enactment of clearly unsupportive coping strategies (ASH) in the face of academic anxiety is likely a response that is prompted by a differential goal set [7,12]. Therefore, rather than adopting a coping strategy that will increase the potential for success on the task (as would be proposed by an academic coach focused on reducing uncertainty and enhancing self-efficacy), students who engage in ASH in the face of CTA are likely focused on reducing the experience of anxiety in the moment [12]. While those strategies improve their affective experience in the moment, the long-term negative effects are demonstrated in progressively lower cumulative GPAs.

### 4.2. Curvilinear Relationships

Research in the field has been dominated by analyses focused on linear relationships among the variables explored in this study. While we believe that the primary finding of this study is demonstrated by the results of the SEM, attention to nonlinear solutions for the relationships among CTA, ASH, and GPA is warranted to continue to broaden understanding of the influence of anxiety and coping on performance among university students [1,15]. For CTA, the relationship demonstrated a strong negative linear relationship with GPA for learners scoring in the moderate range on the CTAS-2. While the slope of the relationship between CTA and ASH maintained an overall negative trend, the slope observed for the low- and high-test anxiety groups was considerably different and indicated a flattening at the two extremes. This indicates greater consistency in predicted GPA for students classified within the low- and high-test anxiety groups, as well as demonstrating significant differences in that performance level.

A similar review of the relationship between ASH and GPA illustrated that there was a bell-shaped curve at the extremely low end of reported ASH. The slight increase in predicted GPA for students who reported engaging in one or two self-handicapping activities infrequently (as compared to those who reported *never* engaging in any self-handicapping activities) was matched with a similar downward trend that led to progressively lower levels of GPA that reached relative stability once learners reported using self-handicapping strategies somewhat regularly.

Finally, examining the influence of the interaction effect of CTA and ASH on GPA provided an opportunity to visualize the serial mediation effect more accurately. Taken together, we believe the curvilinear trends clearly demonstrate that knowledge of a student’s level of CTA or ASH provides some information that can predict cumulative GPA, but greater accuracy and precision are achieved when knowledge of both is available. Specifically, the trends we observed indicate that if students report both high levels of CTA and ASH (upper right corner of Figure 4), the student GPA is predictably low. However, when examining the effects for learners with GPAs in the middle range of our sample, it is clear that a student with high levels of CTA can maintain that GPA, provided the degree of reported self-handicapping is rather low. As the student reports more likelihood of engaging in academic self-handicapping, the CTA level had to drop significantly to maintain a similar GPA (Figure 4, contour lines 3 and 4 from left). Finally, the highest levels of GPA were observed in students who reported both low levels of CTA and ASH (Figure 4, lower left corner).

### 4.3. Limitations and Future Directions

While this study provides useful information for the field by using both a large sample and exploring multiple variables simultaneously, there are areas of limitation to address in continued research. First, the sample is not diverse—with an over-representation of both female and white students from the United States. Continued efforts to diversify the available and participating sample for this work are important. The sample is also focused on a university sample. While we believe that the general patterns would translate to students in traditional secondary schools, the specific effects may vary given that access to university is generally restricted to students with higher overall GPAs. In addition to expanding the age focus in future research, the continuation of this work to samples with a broader range of possible GPAs may also be accompanied by a larger sample of students who self-report academic self-handicapping behaviors. Furthermore, the observational design and statistical methodology of this study (i.e., SEM and curvilinear analysis) are not able to describe causal relationships between intolerance of uncertainty, test anxiety, and grade point average. To determine causality between any of the variables, future research would need to employ an experimental design.

Regarding research and exploration of possible intervention or support efforts, we see several potential future pathways for practitioners and researchers. First, from a methodological stance, we advocate strongly for continued attention to both curvilinear relationships and interactive effects when examining relationships among achievement emotions and outcomes. Our results repeatedly illustrate that the simplistic review of single variables or only linear effects leaves significant room for error in interpretation. Second, from a practical perspective, we see great potential for supporting learners with propensities to academic anxieties. The results of this study support achievement emotion models (e.g., [11,12]) that recognize the importance of knowing the learners’ appraisal of academic contexts, their perceived control over those contexts, and the repertoire of coping strategies at their disposal when navigating those appraisals. Consistent with those models, we advocate for a learner support process that focuses on helping learners accurately identify the task before them as well as the resources and supports at their disposal, helping to reduce the uncertainty that serves as a source of much of their anxiety and subsequent negative outcomes. Further, support in setting achievement goals for specific academic tasks that are focused on managing both achievement emotions and self-regulated learning strategies is essential. When the learner recognizes the duality of managing their emotions (i.e., anxiety, uncertainty) as well as their learning needs (i.e., positive study behaviors), they are more likely to select coping strategies that promote learner engagement and reduce the potential for clearly harmful academic self-handicapping behaviors.

## Figures and Tables

**Figure 1 behavsci-14-00096-f001:**
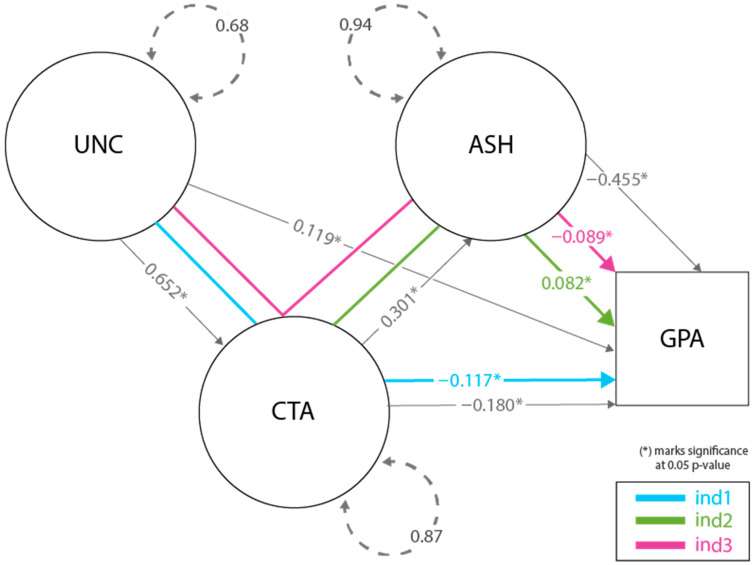
Serial mediation model (Model 4). Factor loadings for individual survey items in Model 4 have been omitted to enhance model clarity and interpretative parsimony. Detailed ranges of these loadings are provided in the main text.

**Figure 2 behavsci-14-00096-f002:**
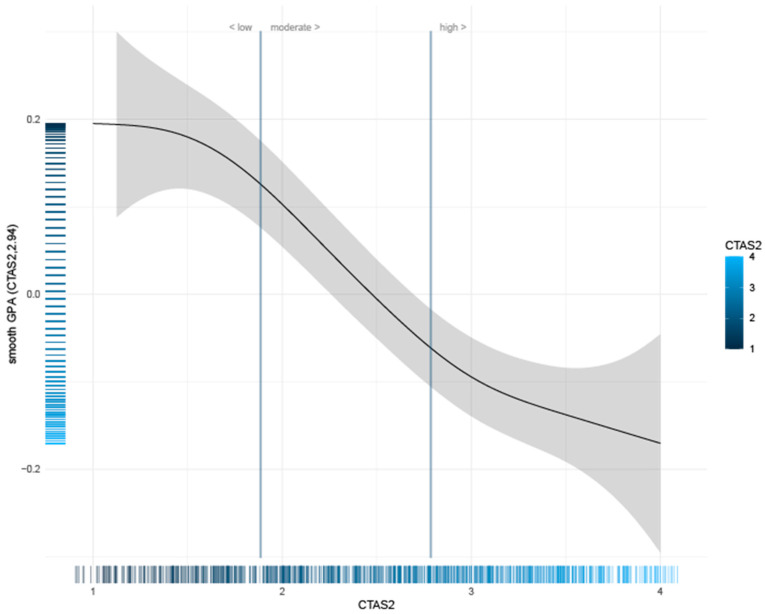
Curvilinear relationship between cognitive test anxiety and GPA.

**Figure 3 behavsci-14-00096-f003:**
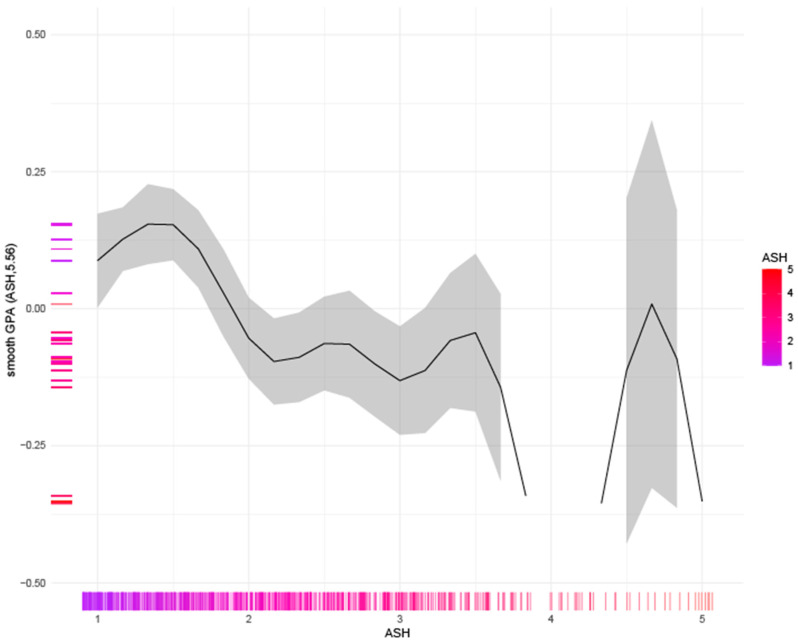
Curvilinear relationship between academic self-handicapping and GPA.

**Figure 4 behavsci-14-00096-f004:**
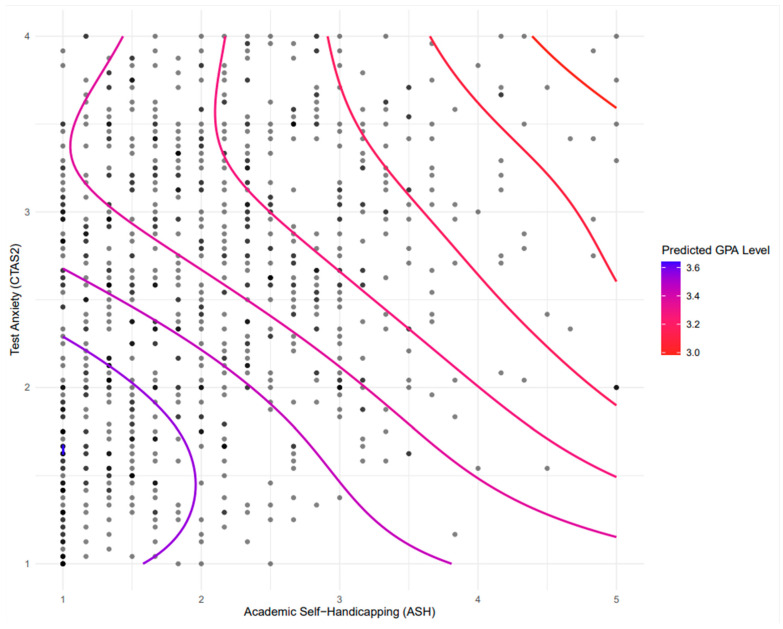
Curvilinear relationships among cognitive test anxiety, academic self-handicapping, and GPA.

**Table 1 behavsci-14-00096-t001:** Demographic distribution.

Variable	
Dimensions	N (%)
Gender	
Female	713 (84)
Male	112 (13)
Non-Binary	19 (2.2)
Other	3 (0.4)
Prefer Not to Respond	3 (0.4)
Not Reported	3
Race	
American Indian or Alaska Native	6 (0.7)
Asian	12 (1.4)
Black or African American	78 (9.2)
Hispanic or Latinx	23 (2.7)
Multiracial	30 (3.5)
Native Hawaiian or Pacific Islander	1 (0.1)
Other	1 (0.1)
White	695 (82)
Not Reported	7
Academic Year	
First Year (Freshman)	208 (24)
Sophomore	228 (27)
Junior	204 (24)
Senior	204 (24)
Graduate School	7 (0.8)
Not Reported	2

**Table 2 behavsci-14-00096-t002:** Model fit comparisons.

Statistic	Model			
	Model 1	Model 2	Model 3	Model 4
*AIC*	87,745.551	599,221.956	87,745.551	87,745.551
*BIC*	88,761.786	601,373.718	88,761.786	88,761.786
*CFI*	0.959	0.276	0.959	0.957
*TLI*	0.95	0.259	0.9	0.947
*RMSEA*	0.04	0.164	0.04	0.04
*SRMR*	0.048	0.229	0.048	0.047
*GPA R* ^2^	0.0937	0.184	0. 104	0.247

Note. AIC = Akaike information criterion; BIC = Bayesian information criterion; aBIC = adjusted Bayesian information criterion; CFI = comparative fit index; TLI = Tucker–Lewis index; RMSEA = root mean square error of approximation; SRMR = standardized root mean square residual; GPA = grade point average.

## Data Availability

Data presented in this study are available upon request from Jerrell C. Cassady (jccassady@bsu.edu), once approval of sharing of the data has been cleared by the local institutional Review Board. Requests for data access must identify requester’s affiliation, intention for data access, and outline procedures for data security. All research instruments used in the study are accessible via cited references, but can also be accessed at https://sites.bsu.edu/aarc (accessed on 20 January 2024).

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
