# Peer review of "The Collective Influence of Intolerance of Uncertainty, Cognitive Test Anxiety, and Academic Self-Handicapping on Learner Outcomes: Evidence for a Process Model"

_behavsci, 2024, doi:10.3390/bs14020096_

Round 1

Reviewer 1 Report

Comments and Suggestions for Authors

Thank you for inviting me to review this paper. See my detailed comments below.

1. Line 26-37, this paragraph lacks reference. For example, line 29, the authors stated “while much research..”, so which previous studies?These should be pointed out.

2. Though this paper has a section of 1.4 current study, it would still be better to add brief sentences to connect the literature to the present study at the end of section 1.1, 1.2, 1.3. E.g., how this study contributes to the current knowledge body.

3. Line 241, the authors said sample is consistent with the university population, so what are the official university’s demographic distributions (i.e., gender, race, and academic year in this study)?

4. the authors deleted the cases that had missing in GPAs. It seems that the authors used the listwise deletion to deal with the missing data. Thus, the final analytic sample size would not be 853. To be more precise, the authors should point out the type of missing data, and identify whether it needs to impute the missingness. Though, I do not believe it is necessary, I would suggest authors to also delete the cases that have missing in gender, race, and academic year, to make the final sample size more clear, and not confuse the readers.

5. line 257, “Intolerance of Uncertainty” or “Uncertainty Intolerance”, it is better to make the same thing consistent in the whole paper.

6. Abbreviations such as CTA, UNC, GPA should be pointed out in the first appearance. Also, please keep the same thing consistent throughout the paper.

7. This is a very long paper, I would suggest the authors to delete Model 2 and not to report its results since it has a poor model fitness, to keep the paper shorter.

8. Figure 1, use solid lines for significant paths, even they are already in colors. GPA is not the latent factor, so use a rectangle instead of a circle. Again, UI or UNC?

9. Since the authors do not report factor loadings in the figures, thus, I would suggest the authors to remove all the survey items in Figure 1.

10. The authors could report the factor loadings range in the main text. For example, the factor loadings for CTA ranged from xxx to xxx, indicating survey items measured CTA well. Also, since there are 12 survey items measuring UI, 24 for CTA, and 6 for ASH, I do not think every item’s factor loading will be high, thus, I think the authors need to adjust their measurement model by deleting low factor loading items, then the model fit indices such as CFI, TLI would be further improved.

11. In the discussion, the authors should add how their findings can have practical implications.

12. the authors should also point out the present study could not draw causal inferences, as one of the limitations.

Minor issues:

1. line 216, I would suggest the authors to change the order to “cognitive test anxiety, uncertainty intolerance, academic self-handicapping” in line with the literature section order. 

Author Response

Reviewer 1:

Thank you for inviting me to review this paper. See my detailed comments below.

  1. Line 26-37, this paragraph lacks reference. For example, line 29, the authors stated “while much research..”, so which previous studies?These should be pointed out.

Thank you for the suggested clarification. We have adjusted the language as well as provided references that identify the “specific assessment materials. We reserve the full explanation of those studies later in section 1.1, but forecasting this more accurately was important.

  1. Though this paper has a section of 1.4 current study, it would still be better to add brief sentences to connect the literature to the present study at the end of section 1.1, 1.2, 1.3. E.g., how this study contributes to the current knowledge body.

Thank you for the suggestion. There was some differential guidance from the review team here, and in order to balance those comments, we have not put summary connections to each section. If the editorial team prefers we follow this reviewer’s recommendation, we will be happy to comply.

  1. Line 241, the authors said sample is consistent with the university population, so what are the official university’s demographic distributions (i.e., gender, race, and academic year in this study)?

This statement was deleted from the paper as part of edits made for this reviewer’s 4th section of recommendations (see below). The consistency of the sample with the university’s population is not necessarily relevant to the overall generalizability of the results.  Demographic information (e.g. race, gender, etc.) on the sample is provided to give readers an ability to critically interpret how representative the sample is to different groups of individuals.

  1. the authors deleted the cases that had missing in GPAs. It seems that the authors used the listwise deletion to deal with the missing data. Thus, the final analytic sample size would not be 853. To be more precise, the authors should point out the type of missing data, and identify whether it needs to impute the missingness. Though, I do not believe it is necessary, I would suggest authors to also delete the cases that have missing in gender, race, and academic year, to make the final sample size more clear, and not confuse the readers.

Revisions of the participant section were made to make both the final sample size, and the reason certain participants were removed more parsimonious. Additionally, a correction was made that now clarifies Table 1 is in fact reflecting the demographic breakdown of the sample used (n=852). Lastly, a more clear statement of the listwise deletion procedure for the relevant participants has been created, as presented in detail below:

“Our final sample consisted of 853 students who completed the survey, which was available from 2021 to 2023. An initial data set of 943 participants was reduced to this sample size after 1) 44 participants were removed as multivariate outliers from a Mahalanobis distance test (df = 42, x2 = 76.3, p = 0.001), and 2) an additional 46 students were removed via listwise deletion due to no reported cumulative GPA. Listwise deletion was selected as the appropriate method for removing participants lacking the single-item outcome variable in our SEM tests, as complete data on this crucial variable is essential for accurate model estimation and the validity of our results. All other missing values of the final sample were subject to imputation. Table 1 reflects the demographic breakdown of our final sample. “ 

  1. line 257, “Intolerance of Uncertainty” or “Uncertainty Intolerance”, it is better to make the same thing consistent in the whole paper.

Switched all occurrences of “Uncertainty Intolerance” to “Intolerance of Uncertainty”, as that is the predominant phrasing used in many of the cited sources. Additionally, this connects with the name of the uncertainty measure used – the Intolerance of Uncertainty 12-item version. In line with this change, instances of the abbreviation of “UI” for “Uncertainty Intolerance” were all changed to “UNC”. This abbreviation was chosen for consistency with the rest of the paper, to address this reviewer’s 6th suggestion.

  1. Abbreviations such as CTA, UNC, GPA should be pointed out in the first appearance. Also, please keep the same thing consistent throughout the paper.

Appropriate parentheticals were added to the introduction paragraph when these concepts are first stated. See response to this reviewer’s 5th recommendation for how consistency in abbreviation was handled of intolerance of uncertainty.

  1. This is a very long paper, I would suggest the authors to delete Model 2 and not to report its results since it has a poor model fitness, to keep the paper shorter.

Thank you for your insightful feedback regarding the length of our paper and the inclusion of Model 2 in our series of Structural Equation Models (SEMs). We acknowledge and appreciate your suggestion to scrutinize the inclusion of a model with poor fitness, typically a prime candidate for removal in efforts to streamline content. However, we maintain that retaining Model 2 in our presentation is crucial for several reasons.

Firstly, our study aims to provide a comprehensive and transparent account of the analytical journey, illustrating the logical progression and evolution of our models. Model 2, despite its poor fit, represents a critical stage in this developmental process. Its inclusion not only demonstrates our methodological rigor but also highlights the iterative nature of model building in SEM analysis. By showing the transition from Model 1 (CFA) through to Model 4, we ensure a complete and clear understanding of how and why certain models were modified or discarded in favor of others.

Model 2, specifically, serves as an essential stepping stone in this narrative. Its failure to demonstrate a significant moderation effect led us directly to the exploration and subsequent confirmation of a mediation effect in Model 3. This progression from a non-significant model to a significant one is a key aspect of our research story, and is indeed a question of significance in the field (other researchers have offered that the primary effect here SHOULD be moderation).

Moreover, Model 2 adheres to the principles of transparency and thoroughness in empirical research. Omitting this model could potentially lead to questions regarding the comprehensiveness of our analysis and the robustness of our conclusions. Our aim is to provide the reader with a full understanding of our analytical process, including both successful and less successful attempts, to enhance the credibility and reliability of our findings.

We feel this is especially crucial given the ethical dangers that a test as powerful and yet unrestricted as SEM can be. In the context of SEM, it is not uncommon for researchers to 'over-fit' models to data, crafting complex relationships that may lack theoretical or empirical grounding. This practice can lead to models that, while statistically adequate, are theoretically unsound or practically implausible. By transparently presenting Model 2, which did not fit well, we are consciously addressing this concern. We demonstrate our commitment to theoretical integrity over mere statistical adequacy, underscoring the importance of aligning model construction with robust theoretical underpinnings and empirical evidence.

Finally – given that the presentation of the model accounts for one short paragraph, we advocate for leaving that section where it is.

  1. Figure 1, use solid lines for significant paths, even they are already in colors. GPA is not the latent factor, so use a rectangle instead of a circle. Again, UI or UNC?

Figure 1 was updated with a version that replaced dashed lines with solid lines in both the figure itself and the figure key. Additionally, survey items were removed for clarity (see item 9). Thank you for the suggestion. We have also adjusted the figure so that the abbreviation for UNC is consistent with the rest of the document.

  1. Since the authors do not report factor loadings in the figures, thus, I would suggest the authors to remove all the survey items in Figure 1.

Figure 1 had all survey items removed and a figure note added that details. This certainly makes the figure less cluttered.

“Factor loadings for individual survey items in Model 4 have been omitted to enhance model clarity and interpretative parsimony. Detailed ranges of these loadings are provided in the main text.”

Factor loading ranges for UNC, CTA, and ASH were placed into the main text (see item 10).

  1. 10. The authors could report the factor loadings range in the main text. For example, the factor loadings for CTA ranged from xxx to xxx, indicating survey items measured CTA well. Also, since there are 12 survey items measuring UI, 24 for CTA, and 6 for ASH, I do not think every item’s factor loading will be high, thus, I think the authors need to adjust their measurement model by deleting low factor loading items, then the model fit indices such as CFI, TLI would be further improved.

Factor loadings for UI (UNC), CTA, and ASH (ranging from lowest to highest) were added to the main body text under section 3.2.1. ‘Model 1: CTA mediating UNC & GPA’ as detailed here:

“Factor loadings for the 24 individual items of CTA ranged from (0.508 - 0.921), the 12 loadings for UNC ranged from (0.431 - 1.038), and 6 item factor loadings for ASH ranged from (0.691 - 1.024).”

Factor loadings are provided in this section, detailing the CFA test fit results, as these loadings will remain consistent throughout each model and only model fit statistics and regression estimates will change from model to model.

  1. In the discussion, the authors should add how their findings can have practical implications.

Following the "future directions" we provide some indication of these practical implications.

  1. the authors should also point out the present study could not draw causal inferences, as one of the limitations.

We have added an additional 2 sentences at the end of the limitations section that highlight this specified limit on the conclusions and implications of this study.

Minor issues

  1. line 216, I would suggest the authors to change the order to “cognitive test anxiety, uncertainty intolerance, academic self-handicapping” in line with the literature section order. 

We agree with this recommendation for consistency and have changed this line’s order accordingly.

Reviewer 2 Report

Comments and Suggestions for Authors

Thank you for the opportunity to review The Collective Influence of Uncertainty Intolerance, Cognitive Test Anxiety, and Academic Self-Handicapping on Learner Outcomes: Evidence for a Process Model. The manuscript was well-written and presented interesting findings. 

The authors thoroughly explore the literature to define the constructs explored as well as what has already been done in the research. If the authors are building on Putwain's study, then this should be highlighted more in the literature. The authors present Pekrun's control theory on page 3. Was this theory important to this particular study? Or do you see this study more conceptually framed around the constructs? This needs to be a bit clearer.

The purpose and hypotheses are clearly stated in the beginning. 

Authors clearly explain how the sample size was obtained. The demographic chart is helpful, though the section on Race is a bit confusing. the n/% column does not seem to align (the N = 227) with the sample size and for black/African American a closed parenthesis is needed.

Authors clearly explain their measures and the tests run to determine model fit. 

Authors present a curvilinear relationship model that provides new insights and a method for exploring the relationships among the constructs that show some validity.

In the discussion, the authors should more clearly compare their findings with what has already been found to make it clear what new contributions they are adding and/or how this work confirms other work that was done. More references to the literature should be woven into the discussion to validate their claims. Additionally, the authors reference how this study builds on Putwain's study but this is only generally discussed on page 5. Was this a goal of the study?  The discussion also does a nice job of evaluating the models which aligns with the study purpose.

Author Response

Reviewer 2:

Thank you for the opportunity to review The Collective Influence of Uncertainty Intolerance, Cognitive Test Anxiety, and Academic Self-Handicapping on Learner Outcomes: Evidence for a Process Model.

The manuscript was well-written and presented interesting findings. 

Thank you for the comment.

The authors thoroughly explore the literature to define the constructs explored as well as what has already been done in the research.

  • If the authors are building on Putwain's study, then this should be highlighted more in the literature. The authors present Pekrun's control theory on page 3. Was this theory important to this particular study? Or do you see this study more conceptually framed around the constructs? This needs to be a bit clearer.

The reviewer is correct that we identify Pekrun’s theory in detail on page 3, but it was also mentioned (without in-text naming) earlier in the process. (section 1.1; citation #12 – we have clarified this, without making Pekrun the “center point” – bc there are similar theories that are consistent and are central to the interpretation by others, including PUtwain’s work you also mention here – he used the S-REF model in one of the studies we focus on – citation 13). That model (along with several others – which are essentially the same core notions – is indeed central to the construct of test anxiety, but was covered in greater detail later (page 3) to tie intolerance uncertainty (which has not been directly related to Pekrun before). Our revision of secion 1.1 (we believe) makes this link more obvious, and hopefully addresses the recommendation.

As for the Putwan reliance, we have attempted to refocus that paragraph, which really is based on 2 studies by Putwain and colleagues (there are several, given Dave’s prolific writing over the past 10 years). Indeed, the two studies central to our study have been described earlier in the appropriate sections (focus on UNC and ASH) – but we have tied these explicitly in the concluding section before the “current study.” We hope the rephrasing clarifies that the primary difference between our work and Putwain’s collection of studies is in the direction or placement of constructs in the tested model.

The purpose and hypotheses are clearly stated in the beginning. 

Authors clearly explain how the sample size was obtained.

  • The demographic chart is helpful, though the section on Race is a bit confusing. the n/% column does not seem to align (the N = 227) with the sample size and for black/African American a closed parenthesis is needed.

Thank you for the comments on the positive information. Also – your observation of the error in that table was much appreciated. These were typographical errors and have been fixed.

Discrepancy and confusion around demographics table is due to the ‘n’ amount for white was not properly typed out, fixed from ‘69’ to 695. Additionally, back parenthesis added for % of ‘Black or African American.’

Authors clearly explain their measures and the tests run to determine model fit. 

Authors present a curvilinear relationship model that provides new insights and a method for exploring the relationships among the constructs that show some validity.

  • In the discussion, the authors should more clearly compare their findings with what has already been found to make it clear what new contributions they are adding and/or how this work confirms other work that was done. More references to the literature should be woven into the discussion to validate their claims.

Again – thank you for the comment about making connections to the field. Direct connection back to prior work (e.g., putwain and pekrun) were inserted.

  • Additionally, the authors reference how this study builds on Putwain's study but this is only generally discussed on page 5. Was this a goal of the study?  The discussion also does a nice job of evaluating the models which aligns with the study purpose.

Also addressed – linked to prior comment as well. We trust the revision, guided by your excellent suggestions, make the paper more useful to the readers.

Reviewer 3 Report

Comments and Suggestions for Authors

a really interesting and well thought out study that demonstrates the non-linear relationships between the various factors relating to test anxiety and performance. This was mostly well explained at clearly reported however it would have been nice to see a little more discussion of the implication of these findings particularly in terms of teaching for those of us who are educators as well as researchers. A theoretical model would also strengthen the paper. One minor error is that some of the literature numbering is incorrect e.g. Putwain, & Pescod.

Comments on the Quality of English Language

Well written with no issues regarding language other than occasional use of slightly repetitive use of phraseology in places.

Author Response

Thank you for your helpful comments - we have reviewed our ordering of the citations as well as clarified the theories you mentioned in the literature review in particular.

Reviewer 4 Report

Comments and Suggestions for Authors

Title and Abstract Review:

The title is clear and informs about the most relevant aspects of the research, a methodological contribution to the field of study of anxiety, of which the authors have experience. However, in the summary, it is necessary to go deeper into the results, focusing on the students' expectations.

Introduction

The introduction is not very clarifying. Although a very interesting and deep conceptual contribution is made, other aspects of the research are scarcely exposed. The epistemological aspects of the research are not mentioned, in the part where the hypothesis is exposed, the motivation of this hypothesis is not clarified in depth. Neither does it state what is the previous literature (line 33) on which they base their research. Hence the lack of epistemological aspects of the research context.

Materials and methods

The Methods section provides a clear description of the study's participants sample, including data collection, variables, and analysis procedures. Overall, it is well-structured and informative. However, there is a suggestion for improvement. It is necessary, as a first point of the methodology, to link this research method to previous ones that have been carried out in the field of study or in other centers. This question will help the reader to situate himself on the methodological design in which the researcher is placed.

Results

The part referring to results indicates the results of the different tools proposed by the author. The Results section of the article presents findings from the study's data analysis, including correlation analysis and mediation analysis. Overall, the presentation is clear, and the results are well-organized. The section is clear and concise in presenting the results. The standardized coefficients and significance levels are reported, making it easy for readers to understand the relationships between variables.

Discussion and conclusions

The Discussion section of the article provides a comprehensive analysis of the study's findings and their implications. It is well-structured and effectively conveys the significance of the research. The discussions, on the other hand, are scarce, and aspects such as the contrast of the hypothesis with the results are not presented.

In summary, I believe that 3 issues should be observed:

1. Improve the introduction of the paper.

2.Expand the methodological part by outlining the research design/approach followed.

3.Revise the section on conclusions.

Author Response

Reviewer 3:

Title and Abstract Review:

The title is clear and informs about the most relevant aspects of the research, a methodological contribution to the field of study of anxiety, of which the authors have experience. However, in the summary, it is necessary to go deeper into the results, focusing on the students' expectations.

We have edited the abstract with your suggestion in mind, specifically focused on addressing the role of the 3 variables in predicting GPA. Naturally, we are limited in the depth we can offer with abstract guidelines. We hope the adjustment fits your intended guidance.

Introduction

The introduction is not very clarifying. Although a very interesting and deep conceptual contribution is made, other aspects of the research are scarcely exposed. The epistemological aspects of the research are not mentioned, in the part where the hypothesis is exposed, the motivation of this hypothesis is not clarified in depth. Neither does it state what is the previous literature (line 33) on which they base their research. Hence the lack of epistemological aspects of the research context.

Our revision focuses largely on this question – making the direct link between our work and the foundational theoretical and empirical work in the field.

Materials and methods

  • The Methods section provides a clear description of the study's participants sample, including data collection, variables, and analysis procedures. Overall, it is well-structured and informative. However, there is a suggestion for improvement. It is necessary, as a first point of the methodology, to link this research method to previous ones that have been carried out in the field of study or in other centers. This question will help the reader to situate himself on the methodological design in which the researcher is placed.

Thank you for your valuable suggestion regarding the enhancement of our methodology section. We agree that establishing a connection between our SEM approach and prior studies in our field will provide readers with a clearer understanding of the methodological context and its evolution. To this end, we will revise our methodology section to include a brief overview of the established process our step-by-step SEM process is situated upon.

"In the development of our methodological approach, particularly our step-by-step Structural Equation Modeling (SEM) process, we draw upon established practices and advancements in the field such as the suggested approach provided by Suhr (2006). Particularly our iterative process detailed in this section highlights the final four suggested steps including parameter estimation (CFA), assessing model fit (Model 1), model modification (Model 2-4), and finally the interpretation and presentation of results. When looking at multiple models of an SEM, these model modifications can be interpreted much in the same way as post-hoc comparisons act for an ANOVA.”

Results

The part referring to results indicates the results of the different tools proposed by the author. The Results section of the article presents findings from the study's data analysis, including correlation analysis and mediation analysis. Overall, the presentation is clear, and the results are well-organized. The section is clear and concise in presenting the results. The standardized coefficients and significance levels are reported, making it easy for readers to understand the relationships between variables.

Thank you for the comments – some revisions have been implemented based on other reviews – but we believe the heart is maintained.  

Discussion and conclusions

The Discussion section of the article provides a comprehensive analysis of the study's findings and their implications. It is well-structured and effectively conveys the significance of the research. The discussions, on the other hand, are scarce, and aspects such as the contrast of the hypothesis with the results are not presented.

We believe your focus here (like one of the other reviewers) is focused on the conclusions/implications. We appreciate the suggestions and hope the revisions have addressed this limitation in the original version.

In summary, I believe that 3 issues should be observed:

  1. Improve the introduction of the paper.

2.Expand the methodological part by outlining the research design/approach followed.

3.Revise the section on conclusions.

Noted – and thank you for the summary. We believe our prior comments have been in line with your recommendations, nothing to add to this.

Round 2

Reviewer 1 Report

Comments and Suggestions for Authors

Thank you for the revision. The manuscript is much improved. Below, I provide additional minor suggestions to tighten up the manuscript even further.

1. Again, the final sample is 853, however, there are missing values in gender, race, and academic year, as shown in Table 1. While the authors used the listwise deletion to deal with the missingness which is fine as I stated in the 1st review, the final ANALYTIC sample size should be less than 853 and is more meaningful to be reported. Thus, I would suggest the authors to remove the missing N in Table 1, and report the final analytic sample size in the text, rather than 853.

2. I may not state clearly in my 1st review. In Figure 1, initially, I would like to suggest the authors to use solid lines for significant results, and dash lines for insignificant results. Then, a note that indicates solid lines for significant results and dash lines for insignificant results would be better added under Figure 1.  

Author Response

Thank you for the continued support in the review - below are your comments, with our response.

  1. Again, the final sample is 853, however, there are missing values in gender, race, and academic year, as shown in Table 1. While the authors used the listwise deletion to deal with the missingness which is fine as I stated in the 1st review, the final ANALYTIC sample size should be less than 853 and is more meaningful to be reported. Thus, I would suggest the authors to remove the missing N in Table 1, and report the final analytic sample size in the text, rather than 853.

You were correct to note that our demographic information included missing subjects in certain domains (which we have relabeled "not reported" rather than missing in Table 1). However, those missing data did not affect our analytic sample. The only subjects who were removed from the dataset were those who were missing the GPA (because it was the DV in the model). Other data were imputed, or not included in the final analytic model - as such, our final reported sample is accurate.

  1. I may not state clearly in my 1st review. In Figure 1, initially, I would like to suggest the authors to use solid lines for significant results, and dash lines for insignificant results. Then, a note that indicates solid lines for significant results and dash lines for insignificant results would be better added under Figure 1.  

Thank you again - we understood your guidance, and believe that the data were reported as expected. All the paths were significant (so, they are all solid lines) - you will see the "dashed" lines here are not actual path lines (so they are not "non-significant" - these are the model estimates (or essentially reliability indicators), and do not have a significant test to report.

Reviewer 4 Report

Comments and Suggestions for Authors

The authors have made substantial modifications throughout the manuscript that improve its quality. However, the paragraph they claim to have included in the methodological section is not detected. It is necessary to include this paragraph since it clarifies the methodological design of the research.

Author Response

We apologize for the missed information - we indeed intended to put it in the section, and failed to do so. It has been repaired - thank you for your attention to this oversight. You will find the paragraph in section 2.3.1, line 316.